# Genomic epidemiology of SARS-CoV-2 in Bolivia, 2020–2024

Esdenka Pérez-Cascales,[1] Francesca Caballero,[2] Anne Martínez-Ventura,[2,3] Brenda Ayzanoa,[2] Mauricio Prado-Zapata,[1] Eliana Baldiviezo-Soliz,[1] Paula Carballo-Jiménez,[4,5,6] Freddy Tinajeros-Guzmán,[4] Diego Cuicapuza,[2,3,7] Pablo Tsukayama[2,8,9]

**ABSTRACT** The COVID-19 pandemic has severely impacted Bolivia due to its high social vulnerability and limited healthcare resources. In response to the limited local capacity for genomic surveillance during the pandemic, we characterized the genetic diversity and geographic distribution of SARS-CoV-2 variants across Bolivia from 2020 to 2024. A total of 714 SARS-CoV-2 genomic sequences were analyzed, including 313 newly sequenced samples and 401 retrieved from GISAID, allowing us to track viral evolution across six epidemic waves. The first wave, dominated by the Wuhan B.1 lineage, resulted in 8,943 deaths, while the fourth wave, characterized by the Omicron BA variant, had the highest infection rates, with approximately 80,000 confirmed cases. While mortality decreased in later waves, case numbers remained high due to the emergence of variants with increased transmissibility and immune escape. Our findings underscore the dynamic evolution of SARS-CoV-2 in Bolivia and highlight the critical need for sustained genomic surveillance to monitor emerging variants and inform public health strategies, particularly in resource-limited settings. Strengthening genomic surveillance, especially in cross-border regions, will mitigate future epidemic waves and improve pandemic preparedness.

**IMPORTANCE** Bolivia's response to the COVID-19 pandemic was challenged by a fragile healthcare system and high social vulnerability. Despite early containment measures, including strict quarantine, the virus spread widely, leading to multiple epidemiological waves that strained the healthcare system and caused significant economic disruption. In this context, genomic surveillance is essential for understanding the evolution of SARS-CoV-2, identifying emerging variants, tracking transmission patterns, and evaluating mitigation strategies. Our study comprehensively analyzes viral dynamics in Bolivia, identifying key transmission hubs and variant replacements across six epidemic waves. These findings underscore the importance of sustained genomic surveillance in resource-limited settings, where real-time monitoring is crucial to anticipate epidemiological trends and enhance pandemic preparedness. Strengthening genomic surveillance capabilities will enhance Bolivia's ability to respond to future health crises and contribute to regional and global pandemic surveillance efforts.

**KEYWORDS** SARS-CoV-2, molecular epidemiology, phylogenetics

The COVID-19 pandemic has been the most severe public health crisis of the 21st century, leading to millions of deaths and overwhelming healthcare systems worldwide (1). Bolivia encountered significant challenges during this emergency, ranking as the most vulnerable country among 31 assessed by the Economic and Social Vulnerability Index from Oxford Economics and Haver Analytics (2). This vulnerability was linked to limited healthcare infrastructure, socioeconomic disparities, and a high reliance on informal labor markets, which hindered the implementation of sustained public health interventions (1, 2).

**Peer Reviewer** Antoni E. Bordoy, Institut d'Investigació en Ciències de la Salut Germans Trias i Pujol, Badalona, Barcelona, Spain

Address correspondence to Pablo Tsukayama, pablo.tsukayama@upch.pe.

The authors declare no conflict of interest.

See the funding table on p. 5.

The first COVID-19 cases in Bolivia were reported on 10 March 2020, in the departments of Oruro and Santa Cruz (3), marking the start of the national epidemic, which intensified by epidemiological week 25 (EW-25) of 2020 (4). On March 12, the Bolivian government introduced its initial response measures to COVID-19, declaring a public health emergency and implementing a strict nationwide lockdown (5). This early intervention successfully reduced the basic reproduction number ($R_0$) from 6 to 2.4 and delayed the epidemic peak by approximately 100 days, helping to prevent an immediate collapse of the healthcare system (6). However, despite these measures, the pandemic continued to spread in multiple waves. By EW-20 of 2023, Bolivia had recorded 1,198,404 confirmed cases and 22,383 deaths, ranking sixth in total cases and seventh in mortality among South American countries (7). Epidemic waves followed a seasonal pattern, with winter surges (June–August in the Southern Hemisphere) primarily affecting Santa Cruz, La Paz, and Cochabamba—Bolivia's most densely populated and economically active regions (7) (Fig. 1).

Early genomic sequencing efforts in 2021 identified two circulating SARS-CoV-2 lineages in Bolivia: B.1.1.348, predominantly found in La Paz and previously reported in Peru, and B.1.1.274, which had not been previously detected in Bolivia but accounted for 44% of cases in the United Kingdom (8, 9). However, Bolivia's genomic surveillance efforts were severely limited, leading to knowledge gaps regarding the evolutionary dynamics of SARS-CoV-2 in the country.

We conducted a genomic study to characterize viral evolution across different epidemic waves. A total of 313 SARS-CoV-2–positive samples, collected between May 2020 and October 2022 in the department of Santa Cruz de la Sierra (the largest in Bolivia by population size), were analyzed using RT-qPCR, with RNA extraction performed via the automated Natch 48 system (Sansure, China). Libraries were prepared using the Illumina COVIDSeq workflow and sequenced on the MiSeq platform, generating 250 bp paired-end reads. Consensus sequences were assembled using the SARS-CoV-2 reference genome (NC_045512.2) through the Illumina DRAGEN COVID Lineage v3.5.3 BaseSpace application. The SARS-CoV-2 sequences were submitted to the GISAID database (EPI_SET_250616sn). To enhance the representation of viral diversity and temporal trends, 401 additional SARS-CoV-2 sequences from Bolivia were retrieved from the GISAID database (11 March 2020 to 27 December 2024). A total of 714 SARS-CoV-2 genomic sequences were analyzed in this study and can be retrieved on GISAID (EPI_SET_250616vh). Sequences were aligned to the Wuhan-Hu-1 reference genome using MAFFT v7.471 (10), and maximum likelihood phylogenies were constructed using IQ-TREE v2.4.0 (11) with the GTR model and 1,000 bootstrap replicates. A time-calibrated phylogenetic tree was generated using TreeTime with a coalescent skyline model, implemented via the augur refine command and the following parameters: --coalescent opt, --clock-rate 0.0008, --clock-std-dev 0.0002, --date-inference marginal, --clock-filter-iqd 4, and --divergence-units mutations. Root-to-tip distances were analyzed using TempEst (12).

By December 2024, Bolivia had experienced six distinct COVID-19 epidemic waves, each dominated by different SARS-CoV-2 lineages. The first wave (March–December 2020) was dominated by the ancestral Wuhan B.1 lineage, resulting in 8,943 deaths, the highest mortality recorded in any wave. The second wave in 2021 was characterized by the importation of Gamma (P.1), first identified in Brazil, while the third wave in mid-2021 saw the displacement of Gamma by the Delta variant (B.1.617.2), which exhibited increased transmissibility. The fourth wave (October 2021–June 2022) was marked by a surge in cases (~80,000 infections), with Omicron BA (BA.1/BA.2) replacing Delta. The fifth and sixth waves (2022–2024) were characterized by the transition from Omicron BA (BA.1/BA.2) to XBB (XBB.1.5) as the dominant lineage (Fig. 2). While mortality decreased over time, case numbers remained high, likely due to the increased transmissibility of later variants and waning immunity in the population. The successive displacement of variants suggests selection for increased transmissibility and immune escape.

Among Bolivia's departments, Chuquisaca was disproportionately affected, recording the highest cumulative incidence (61.4 cases per 100,000 inhabitants) and a case fatality rate of 4%, compared to lower rates in other regions (13). The region has a diversified economic structure, with agriculture and commerce playing important roles—factors that may have contributed to the spread of the virus (14). By late 2020, Mu (B.1.621)

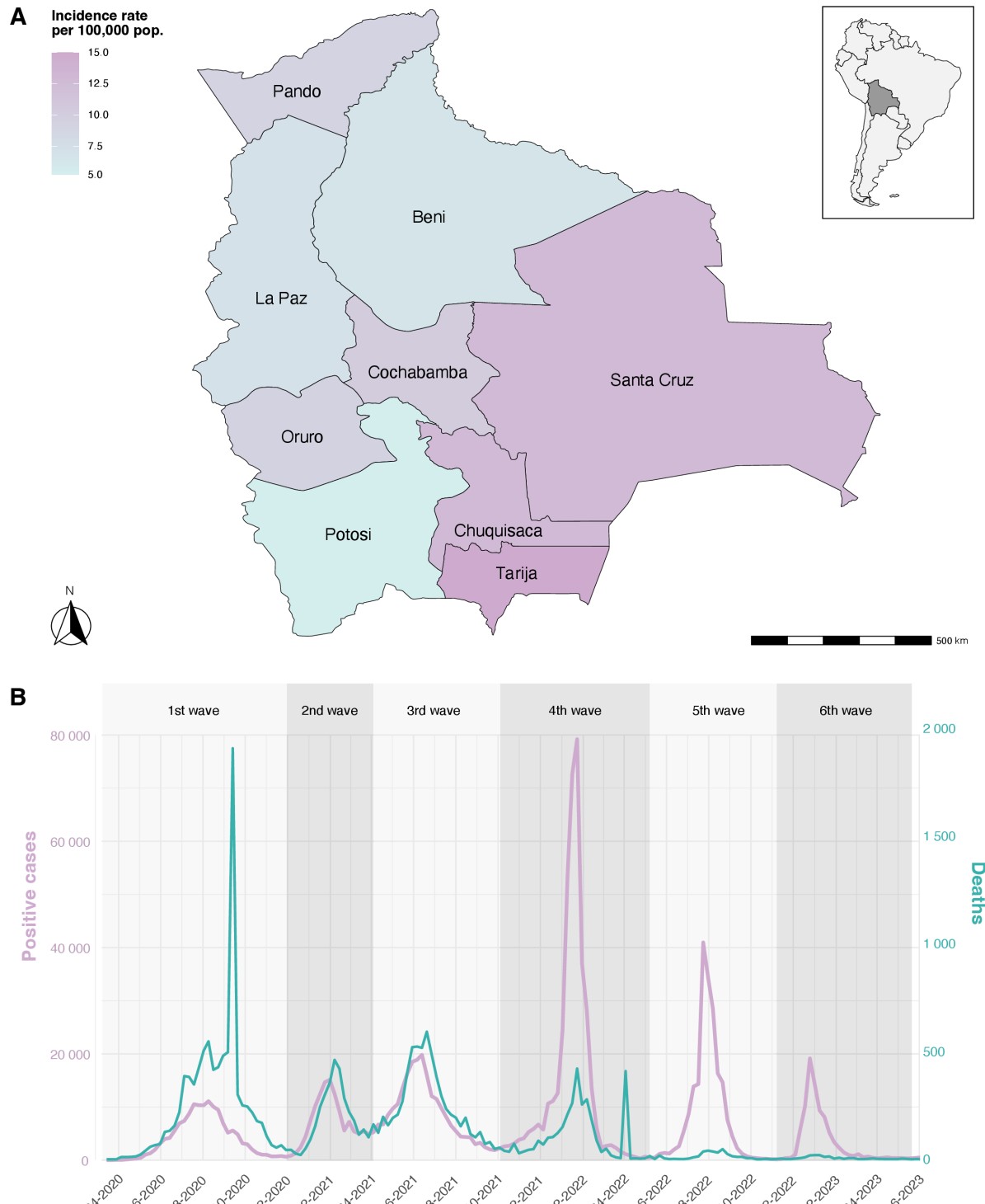

**FIG 1** (A) Incidence rate per 100,000 inhabitants by departments in Bolivia. (B) Case and mortality curve. The daily records of new cases and deaths were obtained from Our World in Data (https://github.com/owid/covid-19-data/tree/master), updated on 19 August 2024. The delineation of epidemiological waves was established according to Ortiz et al. (6).

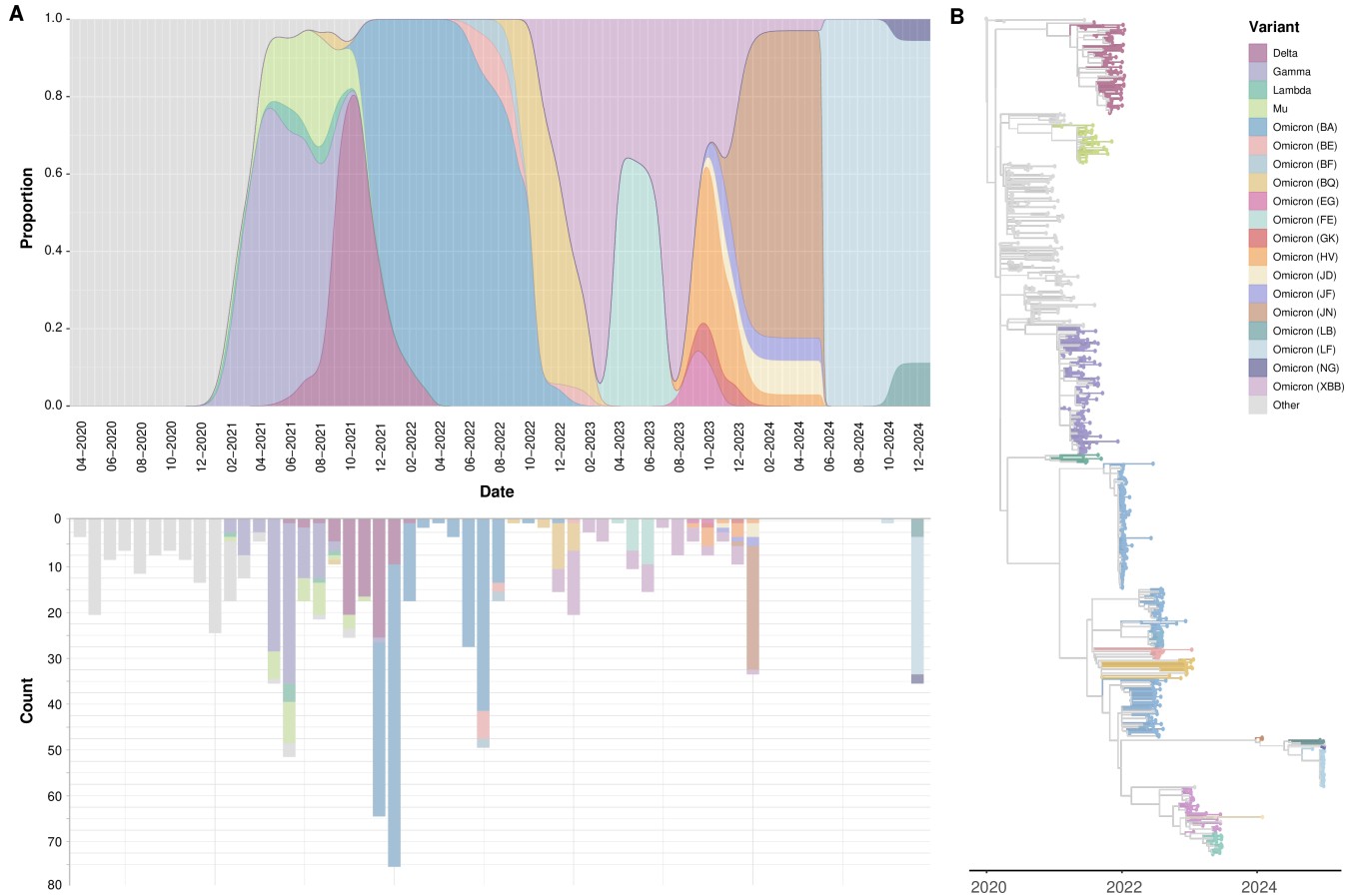

**FIG 2** (A) Top: Relative frequency of SARS-CoV-2 variants. Bottom: Number of sequences shared monthly to GISAID. (B) Time-resolved phylogenetic tree of 675 SARS-CoV-2 isolates.

and Lambda (C.37) (15, 16), first reported in Colombia (17) and Peru (18), were found in Bolivia. However, Lambda did not reach a high prevalence in Bolivia, likely due to competition with co-circulating variants. At the beginning of 2021, the Gamma variant (P.1) had spread rapidly in Bolivia, likely driven by cross-border movement with Brazil and the initially limited availability of vaccines, becoming the predominant variant during the second wave (19, 20). In late 2021, Delta became dominant, aligning with trends observed in other Latin American and Caribbean countries (21). However, by early 2022, Omicron displaced Delta entirely and remained the dominant lineage until the pandemic's decline. This coincided with the fourth wave and the end-of-year holidays, which were associated with increased social mobility and the relaxation of restrictive measures.

This study provides a comprehensive genomic characterization of SARS-CoV-2 in Bolivia (2020–2024). This surveillance effort was implemented using cost-effective strategies, including the use of open-access bioinformatics tools, low-cost sequencing platforms, and regional collaborations that enabled decentralized sample processing and data sharing. These measures highlight a feasible and sustainable model for maintaining genomic surveillance in resource-limited settings such as Bolivia. Enhanced genomic monitoring of cross-border transmission with neighboring countries is essential to mitigate future epidemic waves. These insights can inform public health strategies, improve pandemic preparedness, and support more resilient epidemic wave responses across Latin America.

## ACKNOWLEDGMENTS

This study was funded by Programa Nacional de Investigación Científica y Estudios Avanzados (PROCIENCIA-CONCYTEC) grant PE501086419-2024. A.M.V. and D.C. are supported by a D43 TW007393 training grant awarded to UPCH by the Fogarty International Center of the U.S. National Institutes of Health. The study protocol was approved by the Institutional Review Board at Universidad Peruana Cayetano Heredia (SIDISI 205559).

## AUTHOR AFFILIATIONS

[1]Laboratorio de Diagnóstico e Investigación BIOSCIENCE SRL, Santa Cruz, Bolivia

[2]Laboratorio de Genómica Microbiana, Facultad de Ciencias e Ingeniería, Universidad Peruana Cayetano Heredia, Lima, Peru

[3]Emerge (Emerging Diseases and Climate Change Research Unit), Facultad de Salud Pública y Administración, Universidad Peruana Cayetano Heredia, Lima, Peru

[4]Asociación Benéfica PRISMA, Santa Cruz, Bolivia

[5]Department of Infectious Disease, IFHAD: Innovation For Health And Development, Imperial College London, London, United Kingdom

[6]Department of Tropical Medicine and Infectious Disease, Tulane University, New Orleans, Louisiana, USA

[7]Facultad de Medicina, Universidad Peruana Cayetano Heredia, Lima, Peru

[8]Instituto de Medicina Tropical Alexander von Humboldt, Universidad Peruana Cayetano Heredia, Lima, Peru

[9]Parasites and Microbes Programme, Wellcome Sanger Institute, Hinxton, United Kingdom

## AUTHOR ORCIDs

Francesca Caballero (ID) http://orcid.org/0009-0001-7241-4136
Pablo Tsukayama (ID) http://orcid.org/0000-0002-1669-2553

## FUNDING

| Funder | Grant(s) | Author(s) |
|---|---|---|
| Consejo Nacional de Ciencia, Tecnología e Innovación Tecnológica | PE501086419-2024 | Pablo Tsukayama |
| Fogarty International Center | D43 TW007393 | Anne Martínez-Ventura |
| | | Diego Cuicapuza |

## AUTHOR CONTRIBUTIONS

Esdenka Pérez-Cascales, Conceptualization, Project administration, Validation, Writing – original draft, Writing – review and editing | Francesca Caballero, Data curation, Formal analysis, Investigation, Visualization, Writing – original draft, Writing – review and editing | Anne Martínez-Ventura, Conceptualization, Data curation, Formal analysis, Investigation, Visualization, Writing – original draft, Writing – review and editing | Brenda Ayzanoa, Conceptualization, Data curation, Formal analysis, Supervision, Writing – original draft, Writing – review and editing | Mauricio Prado-Zapata, Conceptualization, Resources | Eliana Baldiviezo-Soliz, Conceptualization, Resources | Paula Carballo-Jiménez, Conceptualization, Methodology, Writing – review and editing | Freddy Tinajeros-Guzmán, Conceptualization, Validation, Writing – review and editing | Diego Cuicapuza, Conceptualization, Data curation, Formal analysis, Methodology, Validation, Visualization, Writing – original draft, Writing – review and editing | Pablo Tsukayama, Conceptualization, Data curation, Funding acquisition, Investigation, Methodology, Supervision, Validation, Writing – original draft, Writing – review and editing

## ADDITIONAL FILES

The following material is available online.

### Open Peer Review

**PEER REVIEW HISTORY (review-history.pdf).** An accounting of the reviewer comments and feedback.

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
