## [Reviewer comments · Microbiology Spectrum]

Microbiology Spectrum

GENOMIC EPIDEMIOLOGY OF SARS-COV-2 IN BOLIVIA, 2020-2024

Esdenka Pérez-Cascales, Francesca Caballero, Anne Martinez-Ventura, Brenda Ayzanoa, Mauricio Prado-Zapata, Eliana Baldiviezo-Soliz, Paula Carballo-Jimenez, Freddy Tinajeros, Diego Cuicapuza, and Pablo Tsukayama

Corresponding Author(s): Pablo Tsukayama, Universidad Peruana Cayetano Heredia

Review Timeline:

Submission Date:	April 23, 2025
Editorial Decision:	June 15, 2025
Revision Received:	June 23, 2025
Accepted:	June 29, 2025

Editor: Daniel Ortiz

Reviewer(s): Disclosure of reviewer identity is with reference to reviewer comments included in decision letter(s). The following individuals involved in review of your submission have agreed to reveal their identity: Antoni E. Bordoy (Reviewer #1)

Transaction Report:

DOI: <https://doi.org/10.1128/spectrum.01280-25>

Re: Spectrum01280-25 (**GENOMIC EPIDEMIOLOGY OF SARS-COV-2 IN BOLIVIA, 2020-2024**)

Dear Dr. Pablo Tsukayama:

Thank you for the privilege of reviewing your work. Below you will find instructions from the Spectrum editorial office and the reviewer comments.

Revision Guidelines

Sincerely,
Daniel Ortiz
Editor
Microbiology Spectrum

Reviewer #1 (Comments for the Author):

Minor comments:

Please state that sequences obtained in Santa Cruz de la Sierra by the authors were submitted to GISAID. If possible, provide an EPI_SET ID of all sequences analyzed in the study.

Line 242. Figure 1 caption. Please change "cities in Bolivia" to "Departmens of Bolivia" if appropriate.

Figure 2. Please change "Otros" to "Other" in the legend.

Reviewer #2 (Comments for the Author):

This study systematically analyzes the genomic evolutionary dynamics of SARS-CoV-2 in the six waves of COVID-19 in Bolivia from 2020 to 2024, integrated 313 new sequencing data and 401 public data, revealed the variant replacement patterns (e.g., Gamma cross-border transmission, Omicron dominated late epidemic), identified key transmission centers and variant substitutions in the six epidemic waves, and pointed out the high infection rate in the Chuquisaca region and the importance of cross-border surveillance.

In this study, IQ-TREE and GTR models were constructed, and technologies such as TreeTime and TempEst were used to fill the gap of genomic surveillance in resource-limited areas, providing key data and data guarantee for regional prevention and control. The structure of the paper is clear, and the data is abundant, but some of the methodological details and the depth of the conclusions need to be strengthened.

1. What are the specific parameter settings of the 'coalescent skyline model' in TreeTime?

2 How does Figure 2a (variant frequency time series) illustrate the association between key public health events and variant change?

3 The paper emphasizes the importance of surveillance under resource constraints, but does not propose a low-cost solution, and the possibility of sustainability of genomic surveillance?

4 The full text of "epidemic wave" and "outbreak" is mixed, and it is recommended to unify them as "epidemic wave". Reference format: Some entries have incomplete information (e.g., references 4 and 5 are missing journal titles and publishers), which need to be completed according to the journal's requirements.

The study provides a comprehensive analysis of the genetic diversity and distribution of SARS-CoV-2 variants in Bolivia over a four-year period that is suited for an "Observation" article. The study analyzed a total of 714 genomic sequences, including 313 newly sequenced samples and 401 retrieved from the GISAID database, to track the evolution of SARS-CoV-2 across four years and six epidemic waves.

The findings provide good quality data for future reference of SARS-COV-2 genetic variants predominance in Bolivia. Thus, they are of great interest to the broad microbiology community of South America.

Minor comments:

Please state that sequences obtained in Santa Cruz de la Sierra by the authors were submitted to GISAID. If possible, provide an EPI_SET ID of all sequences analyzed in the study.

Line 242. Figure 1 caption. Please change "cities in Bolivia" to "Departments of Bolivia" if appropriate.

Figure 2. Please change "Otros" to "Other" in the legend.

Spectrum01280-25: Response to Reviewers

We thank the reviewers and the editorial team for their constructive comments. Below, we provide point-by-point responses to each suggestion, and describe the corresponding revisions made to the manuscript.

Reviewer #1:

- 1. Please state that the sequences obtained in Santa Cruz de la Sierra by the authors were submitted to GISAID. If possible, provide an EPI_SET ID of all sequences analyzed in the study.**

We have added a statement to the manuscript confirming that all 313 sequences generated from Santa Cruz de la Sierra were submitted to GISAID. Additionally, we now include the GISAID dataset ID EPI_SET_250616vh, which comprises all 714 sequences analyzed in this study (both newly generated and publicly available).

- 2. Line 242. Figure 1 caption. Please change "cities in Bolivia" to "Departments of Bolivia" if appropriate.**

This has been corrected in the revised figure caption, which now reads “Departments of Bolivia.”

- 3. Figure 2. Please change "Otros" to "Other" in the legend.**

The legend label “Otros” has been translated to “Other” in the updated figure.

Reviewer #2:

This study systematically analyzes the genomic evolutionary dynamics of SARS-CoV-2 in the six waves of COVID-19 in Bolivia from 2020 to 2024, integrated 313 new sequencing data and 401 public data, revealed the variant replacement patterns (e.g., Gamma cross-border transmission, Omicron dominated late epidemic), identified key transmission centers and variant substitutions in the six epidemic waves, and pointed out the high infection rate in the Chuquisaca region and the importance of cross-border surveillance.

In this study, IQ-TREE and GTR models were constructed, and technologies such as TreeTime and TempEst were used to fill the gap of genomic surveillance in resource-limited areas, providing key data and data guarantee for regional prevention and control. The structure of the paper is clear, and the data is abundant, but some of the methodological details and the depth of the conclusions need to be strengthened.

1. What are the specific parameter settings of the 'coalescent skyline model' in TreeTime?

We have now explicitly listed the parameters used for TreeTime's coalescent skyline model in the Methods section. These include:

--coalescent opt, --clock-rate 0.0008, --clock-std-dev 0.0002, --date-inference marginal, --clock-filter-
iqd 4, and --divergence-units mutations.

2. How does Figure 2a (variant frequency time series) illustrate the association between key public health events and variant change?

We have revised the manuscript text to better articulate how Figure 2a indirectly reflects the influence of public health events—such as vaccination rollout, seasonal holidays, and changes in mobility restrictions—on variant dynamics. While these events are not directly annotated in the figure, the revised discussion explicitly connects epidemiological trends with shifts in variant prevalence.

3. The paper emphasizes the importance of surveillance under resource constraints, but does not propose a low-cost solution, and the possibility of sustainability of genomic surveillance?

We agree with this important observation. In response, we have expanded the final paragraph of the manuscript to explicitly describe the cost-effective strategies employed in this study—such as the use of MiSeq sequencing, open-source bioinformatics tools, and decentralized data analysis via regional

collaborations. We also discuss the sustainability of this model for genomic surveillance in resource-limited settings.

- 4. The full text mixes the terms “epidemic wave” and “outbreak.” Please unify terminology.**

We have reviewed the manuscript thoroughly and replaced all instances of “outbreak” with “epidemic wave” to ensure consistency throughout the text.

- 5. Reference format: Some entries have incomplete information (e.g., references 4 and 5 are missing journal titles and publishers), which need to be completed according to the journal's requirements.**

Reference 4 has been updated to include complete bibliographic details. Reference 5, an official guideline from the Ministry of Health of Bolivia, has been clarified as a government publication and cited following ASM style guidelines.

Re: Spectrum01280-25R1 (**GENOMIC EPIDEMIOLOGY OF SARS-COV-2 IN BOLIVIA, 2020-2024**)

Dear Dr. Pablo Tsukayama:

Your manuscript has been accepted, and I am forwarding it to the ASM production staff for publication. Your paper will first be checked to make sure all elements meet the technical requirements. ASM staff will contact you if anything needs to be revised before copyediting and production can begin. Otherwise, you will be notified when your proofs are ready to be viewed.

Sincerely,
Daniel Ortiz
Editor
Microbiology Spectrum